# Indoor Air Quality Evaluation Using Mechanical Ventilation and Portable Air Purifiers in an Academic Dentistry Clinic during the COVID-19 Pandemic in Greece

**DOI:** 10.3390/ijerph18168886

**Published:** 2021-08-23

**Authors:** Ioannis Tzoutzas, Helena C. Maltezou, Nikolaos Barmparesos, Panagiotis Tasios, Chrysanthi Efthymiou, Margarita Niki Assimakopoulos, Maria Tseroni, Rengina Vorou, Fotios Tzermpos, Maria Antoniadou, Vassilios Panis, Phoebus Madianos

**Affiliations:** 1School of Dentistry, National and Kapodistrian University of Athens, 11527 Athens, Greece; tzoudent@dent.uoa.gr (I.T.); fhtomfs1961@gmail.com (F.T.); mantonia@dent.uoa.gr (M.A.); vpanis@dent.uoa.gr (V.P.); pmadian@dent.uoa.gr (P.M.); 2Directorate of Research, Studies, and Documentation, National Public Health Organization, 15123 Athens, Greece; 3Department of Applied Physics, Faculty of Physics, National and Kapodistrian University of Athens, 15772 Athens, Greece; nikobar@phys.uoa.gr (N.B.); panagiotistasios@gmail.com (P.T.); c-efthymiou@phys.uoa.gr (C.E.); masim@phys.uoa.gr (M.N.A.); 4Directorate of Epidemiological Surveillance for Infectious Diseases, National Public Health Organization, 15123 Athens, Greece; m.tseroni@eody.gov.gr (M.T.); rvorou@gmail.com (R.V.)

**Keywords:** dentistry school, COVID-19, indoor air quality, purifiers, ventilation

## Abstract

The coronavirus disease 2019 (COVID-19), which is mainly transmitted through droplets without overlooking other sources of transmission, rendered attention on the air quality in indoor areas and more specifically in healthcare settings. The improvement of indoor air quality (IAQ) is ensured by frequent changes of the air that must be carried out in healthcare areas and with assistance from special devices that undertake the filtration of the air and its purification through special filters and lamps. In this research, the performance of air purifiers is assessed in terms of the limitation of PM_2.5_, PM_10_, VOCs and CO_2_ in a postgraduate clinic of the Dentistry School of the National and Kapodistrian University of Athens in parallel with mechanical ventilation. Our findings indicate that the use of mechanical ventilation plays a key role on the results, retaining good IAQ levels within the clinic and that air purifiers show a positive impact on IAQ by mainly reducing the levels of PM_2.5_ and secondly of TVOC.

## 1. Introduction

For more than a year and a half, the world has been faced with a pandemic caused by a new coronavirus named severe acute respiratory syndrome coronavirus 2 (SARS-CoV-2), the etiologic agent of coronavirus disease 2019 (COVID-19) [1]. SARS-CoV-2 is easily transmitted mainly through respiratory droplets [2] and it can remain viable in aerosols, depending on the temperature and humidity [3]. During the first pandemic wave and before the transmission dynamics of the novel coronavirus were known, high SARS-CoV-2 RNA concentrations were detected in a few medical personnel areas in two Chinese hospitals but were negligible after meticulous cleaning and disinfection; in contrast, low concentrations of viral RNA were found in aerosols in isolation rooms and ventilated patient rooms [4]. Another experimental study found that SARS-CoV-2 maintains infectivity at a respirable particle size over short distances and virion integrity for up to 16 h [5]. Moreover, the role of air-conditioning in the transmission of SARS-CoV-2-contaminated respiratory droplets via the airflow was also shown in three COVID-19 outbreaks [6] and potentially in a nosocomial outbreak leading to a superspreading event [7]. In this context, dental healthcare facilities, including dental schools, have several characteristics that may facilitate the spread of SARS-CoV-2 among patients, academic personnel, supportive personnel and dental students. These include: firstly, close face-to-face contact with the patients and, in particular, with their oral and upper respiratory tract secretions; second, the aerosol-generating procedures (AGPs) that are commonly applied in dental healthcare settings leading to the generation of droplets 0.5 to 5 micrometers (μm) in diameter that can remain suspended based on the environmental conditions; third, the high risk of virus transmission through the AGPs; fourth, the concomitant implementation of dental procedures in the same practice area and fifth, the ability of both asymptomatic and symptomatic SARS-CoV-2-infected patients to transmit the virus [8]. Virus-contaminated droplets generated during the AGPs may be inhaled and infect dentists, staff and patients. The latter should wear an FFP3 respirator, goggles, a waterproof protective gown and gloves during the AGPs [8].

In the past decade, studies have been conducted to assess the indoor air quality (IAQ) in various settings including healthcare facilities where the mixture of chemical pollutants forms health-burdened indoor conditions for both patients and healthcare personnel [9]. The implementation of a heating ventilation and air-conditioning (HVAC) system under specific requirements is imperative to ensure an adequate IAQ within dental clinics [8,10]. In dental settings where the HVAC system has no high-efficiency particulate absorbance/air (HEPA) filters, mobile air filtering units can be used close to the dental healthcare areas, considering the room size, the number of people and time spent in the room, the air changes per hour (ACH) and the characteristics of the aerosols produced [8]. It is estimated that the potential infectiousness of the air is reduced from 100% to <1% within one hour for 6 ACH and within 30 min for 10–12 ACH [11,12].

In the context of the ongoing COVID-19 pandemic, studies were recently conducted to investigate the efficiency of portable air purifiers in homes and various workplaces [13,14,15]. However, to the best of our knowledge, there is no published study investigating the efficiency of portable air purifiers in academic dental clinics.

The purpose of this study was to evaluate the performance of portable HEPA air purifiers to improve the IAQ of an academic institution dentistry clinic.

## 2. Material and Methods

### 2.1. Setting

The study was conducted in the Faculty of Dentistry of the National and Kapodistrian University of Athens (Athens, Greece). The Endodontics, Periodontics and Operative Dentistry Clinic (hereafter referred as “the clinic”) where the study was conducted is located on the second floor of the main postgraduate building and expands to a space of 170 m^2^ (510 m^3^). The clinic operates in two shifts. The morning shift is from 08:30 a.m. to 12:25 p.m. with approximately 27–35 people (staff and patients) and the noon shift is from 13:30 p.m. to 16:45 p.m. with approximately 16–29 people.

The study period comprised four monitoring periods: from 29 April–16 May 2021, to obtain background data (from 29 April–9 May 2021 when no personnel attended the clinic and from 10–16 May 2021 when only security staff visited the clinic) and 17–21 May, 24–28 May and 31 May–4 June 2021 when the medical staff worked under different operational patterns of the air purifiers.

### 2.2. Ventilation Equipment

The existing air-conditioning/ventilation equipment of the clinic includes two 6 Kw split units of a cooling capacity, which can cover the cooling load of the ventilation system. When occupied, the ventilation system is used and constantly infiltrates fresh air within the room. Τo investigate the impact of the two air purifiers on the IAQ levels of the clinic, various operational patterns for the different shifts during the experimental period were selected (Table 1).

The performance of the Aurabeat AG+ NSP-X1 air purifiers (from now on named P1 and P2) installed within the clinic was also evaluated. According to the manufacturers, the purifiers contain filters that can mitigate the levels of total volatile organic compounds (TVOC) and particulate matter 2.5 (PM_2.5_), which are frequently related to viral loads such as COVID-19 [16]. In Figure 1, the experimental points of the IAQ TSP-18 sensors within the clinic are depicted. The positions of the instruments were selected considering the following criteria: first, targeting tο sufficiently cover the net area of the clinic (approximately 150 m^2^); second, in order to maintain adequate distances from the air purifiers, two sensors were positioned at a distance of less than 2 m from each purifier and two others further than 4 m from each one of them; and third, in order to mitigate the bias from the infiltration of external fresh air, the IAQ TSP-18 sensors were situated as far as possible from window openings. Two sensors were positioned close to the working benches and two others next to the main corridor.

Figure 2a,b illustrates an IAQ TSP-18 sensor along with the mechanical ventilation system within the clinic.

### 2.3. Interventions

The minimum required ventilation with 6 ACH is 3.060 m^3^/h (510 m^3^ × 6 ACH). A new network of vents (through the false ceiling) and orifices in the roof were installed to improve the air dissipation near the workstations, which was able to discard 3.600 m^3^/h with 7 ACH. Due to the aerosol production emitted from the various air/water cooled high-speed rotative cutting and scaling instruments, portable HEPA units (P1 and P2 air purifiers) were installed in areas where the AGPs were performed. Four portable Tongdy TSP-18 real-time output of CO_2_, temperature and relative humidity (RH) sensors designed for wireless connection via a cloud were placed within the clinic to adequately cover the volume of the room (510 m^3^); each one of them was approximately at a 1 m height from the ground. All instruments were in conformity with the GB/T19001-2016/ISO9001:2015 standard.

### 2.4. Data Collection

All data were recorded continuously at 1 min intervals on a 24 h basis and were uploaded onto an online platform with the use of a wireless local area network (WLAN). The sensors performed simultaneous measurements of the temperature and RH levels along with concentrations of CO_2_, TVOC, PM_2.5_ and PM_10_.

The PM sensor operated with a light scattering method (laser particle sensor) and had a recording range for both PM_10_ and PM_2.5_ from 0 to 500 μg/m^3^ with an accuracy of <±15%. The CO_2_ sensor used a non-dispersive infrared detector (NDIR) and ranged from 400 to 2000 ppm, with an accuracy of ±75 ppm. The TVOC sensor ranged from 0 to 4 mg/m^3^ with an accuracy of ≤±0.05 mg/m^3^ based on the manufacturer’s TVOC module. In addition, a high-precision digital integrated sensor measured the temperature from −20 to 60 °C with an accuracy of <±0.5 °C and the RH from 0 to 99% with an accuracy of <±3%. Finally, a logbook of activities was kept recording the different human presence patterns within the room, the operating hours of the air purifiers, any possible door or window opening and other useful information. A five-week monitoring protocol was scheduled and IAQ measurements were initiated on 29 April 2021, when no educational activities took place for 15 days, to monitor the threshold concentrations of CO_2_, PM_2.5_, PM_10_ and TVOC (background data).

## 3. Results and Discussion

### 3.1. Overall Monitoring

Figure 3 presents the levels of the RH and temperature within the clinic throughout the study period. Table 2 illustrates the results from all four IAQ sensors during the study period. The RH and temperature levels remained relatively stable throughout the study period and within the respective limits set by the American Society of Heating, Refrigerating and Air-Conditioning Engineers (ASHRAE) [17] for most observations. More specifically, >95% of RH measurements were found to be within the range of 30–60% and about 70% of the temperature was within 23–26 °C. Thus, the thermal conditions within the clinic were adequate. The levels of all investigated air pollutants were low except for TVOC. Higher levels of TVOC were expected during operational hours as the frequent use of dental materials and biomaterials (such as dentin bonding agents, acrylic resin materials for temporary restorations, various primers and disinfection materials) may lead to extremely high TVOC emissions. More specifically, PM_10_ and PM_2.5_ presented relatively low concentrations from 2.05 to 72.63 μg/m^3^ with an average of 6.80 μg/m^3^ and from 1.58 to 71.40 μg/m^3^ (mean 6.30 μg/m^3^), respectively. The levels of CO_2_ were recorded from 375 to 1407 ppm (average 457 ppm) and concentrations of TVOC ranged from 0.03 to 4.00 mg/m^3^ with an average value of 0.79 mg/m^3^.

Figure 4 shows the concentrations of CO_2_ in the clinic. Only seven outlier values of CO_2_ (<1% of the data) exceeded the threshold limit of 1000 ppm as set by Eykelbosh [18]. Indoor CO_2_ is associated with human exhalation and, as expected, was higher when occupancy within the clinic was elevated (24–30 May 2021 and 31 May–4 June 2021). Indeed, compared to the week prior to the start of the clinic’s operations (10–16 May 2021), CO_2_ increased slightly (4%) from 17–23 May 2021 when the clinic was fully operational. This could be attributed to the full operation of the mechanical ventilation system installed within the clinic especially during the morning shifts when the room was more crowded. Thus, the ventilation levels of the room were found to be sufficient even when the occupancy levels were elevated. It should be noted that air purifiers do not influence the concentrations of CO_2_. These results were in compliance with the study of Meyer [19]. In the latter study, the researchers reported that the CO_2_ concentration levels in a dentistry clinic are associated with the number of occupants and the ventilation conditions, which can reach the highest values when the maximum number of people occupy the clinic and insufficient air renewal is noted.

Figure 5 shows the concentrations of TVOC in the clinic. As expected, the concentrations of TVOC started to increase during the second week prior the opening of the clinic (10–16 May 2021), probably due to the use of detergents from the cleaning services. Regarding the working days, TVOC showed an average concentration of 0.75 mg/m^3^ during the first week of operations (17–23 May 2021), increasing during the following two weeks with average values of 1.3 mg/m^3^ (24–30 May 2021) and 1.5 mg/m^3^ (31 May–4 June 2021), respectively. Considering the dental materials used in the clinic, a few instant maximum TVOC concentrations were expected to be observed that surpassed the threshold limits proposed by Helmis et al. (0–0.3 mg/m^3^ very good, 0.3–1 mg/m^3^ good, 1–3 mg/m^3^ medium, >3 mg/m^3^ poor) [9]. Nonetheless, more than 75% of the recorded values were measured between 0 and 1 mg/m^3^, indicating relatively satisfying IAQ levels. Similarly, according to another study conducted by Helmis et al., the main problem of dental clinics is the TVOC emissions [20]. In this study, the researchers annotated very high concentrations of TVOC associated with the nature of the dental activities and the ventilation conditions [20].

Concentrations of PM_2.5_ were significantly low during the whole experimental period except for a few cases that demonstrated outlier values (Figure 6). Those cases were observed during the background measurements (29 April–9 May 2021) and amid the first operational week of the clinic (17–23 May 2021). For the first case, the higher concentrations of PM_2.5_ may be attributed to barbeque activities around the area due to Easter festivities (2 May 2021). The second case was mainly associated with a massive fire that broke out at a forest nearby area (Corinth) from 19–20 May 2021, approximately 70–80 km away from the clinic. Added to that, strong winds transferred large amounts of smoke over the Athens region, increasing the concentration of local particulate matter. It should be noted that for both cases a few windows were open within the room and, thus, outdoor unfiltered air significantly affected the results. For the remaining cases, PM_2.5_ remained at very low levels considering the operations that took place during the two working shifts and never surpassed the daily standard exposure limit of 25 μg/m^3^ as proposed by the World Health Organization [21]. This result highlights the key role of the mechanical ventilation system installed within the room. During working hours, PM_2.5_ values ranged from 1.75 μg/m^3^ to 11.4 μg/m^3^ excluding the cases that were related to the fires on 20 May 2021. During the first week of operation (17–23 May 2021), the lowest concentrations of PM_2.5_ were observed whereas during the following two weeks, the levels of PM_2.5_ increased by 133% (24–30 May 2021) and 118% (31 May–4 June 2021), respectively. PM_10_ followed mainly the same pattern. Other authors have found high concentrations of particulate matter in dental clinics, which were attributed to the special materials used and to the dental procedures and treatments performed [9]. External pollution seemed to have a negligible contribution. However, the occupancy, as well as the efficiency of the ventilation system used, affects the final measured concentration [22]. Avoiding the recycling of unfiltered air is very important mainly in healthcare-related installations as the airborne transmission of the virus is possible, as mentioned above [14,15,23,24]. In a recent study, it was shown that the combination of the intraoral and extraoral mechanical suction of the produced aerosol had better results in reducing the microbial load in the air than the use of intraoral alone [25]. The same conclusions arose in a very recent article stating that even portable high-efficiency particulate filters may be installed especially in an old-fashioned ward in healthcare institutions with suboptimal ventilation [13].

In Table 3, which refers to the morning shifts, it is clear that from the first operational week (17–21 May 2021), when both air purifiers were activated, there was a significant increase in all examined air pollutants during 24–28 May 2021 and 31 May–4 June 2021. More specifically, from 24 May–28 May 2021 when P1 was off and P2 was operating, an increase of 40% inPM_10_, 53% inPM_2.5_ and 170% in TVOC was observed. Similarly, from 31 May–4 June 2021 when both P1 and P2 were off, PM_10_ increased by 49%, PM_2.5_ by 64% and TVOC by 151%. Interestingly, TVOC demonstrated a lower increase in rate when both air purifiers were not operating (151%) compared to the increase when P2 was switched on (170%). However, it should be noted that elevated concentrations of TVOC are, as mentioned, highly associated with the frequent use of dental biomaterials and not only from the operation of an air purifier.

In Table 4, concerning the noon shifts (with lower levels of occupancy), it can be seen that the operation of P2 led to improved levels of IAQ within the clinic. The week when only P2 was operating (17–21 May 2021) was compared with 24–28 May 2021 when both purifiers were switched off and 31 May–4 June 2021 when only P1 was operating. It is obvious that, from 24–28 May 2021, there was an increase of 170% for PM_10_, 202% for PM_2.5_ and 226% for TVOC in relation to17–21 May 2021. Additionally, from 31 May–4 June 2021, PM_10_ increased by 129%, PM_2.5_ by153% and TVOC by 234% in relation to 17–21 May 2021. Thus, P2 demonstrated better results than P1 for all examined air pollutants.

### 3.2. Days of Interest

In order to further investigate the impact of the air purifiers on the IAQ levels of the clinic, days of interest from each operational week were selected as case studies and are illustrated in Figure 7 and Figure 8. Figure 7 depicts that on 21 May 2021, the hourly concentrations of PM_2.5_ remained very low during the morning shift when both P1 and P2 were switched on and slightly increased amid the noon shift when only P2 was operating within the room. Another interesting result is that during the morning shift of 28 May 2021, the operation of P2 significantly reduced the concentrations of PM_2.5_, which increased amid the noon shift when both purifiers were switched off. On 3 June 2021, when the morning shift started (08:30 a.m.) with both P1 and P2 deactivated, the concentrations of PM_2.5_ increased sharply. After the activation of P1, the concentrations decreased significantly, as expected. For all cases, an unexpected increase in PM_2.5_ during the nighttime (after 21:00 p.m.) may be attributed to external air infiltrating into the room from opened windows. The behavior of TVOC was found to be relatively similar to PM_2.5_ with an exception for 28 May 2021. Figure 8 illustrates that, during the morning shift of 21 May 2021, the operation of both air purifiers managed to retain low levels of TVOC within the room. Those levels slightly increased amid the noon shift when only P2 was activated, as expected. On 28 May 2021, the operation of P2 did not manage to decrease the concentrations of TVOC during the morning shift, probably due to the frequent use of dental materials on this day. Interestingly, even if both purifiers were deactivated on the noon shift, the levels of TVOC were slightly decreased. Finally, on 3 June 2021, when P1 and P2 were switched off amid the morning shift, TVOC was found to be elevated until the end of the shift (12:45 p.m.) followed by a slight decrease when P1 was initiated for the noon shift.

In general, the deployment of the two purifiers showed positive results on the purification of the clinic’s indoor environment when operating simultaneously. However, the improvement was clear mainly for PM_2.5_ as TVOC was highly dependent on the dental materials that were used during the different days and working shifts.

## 4. Conclusions

In situ measurements of IAQ and thermal conditions were conducted in a dentistry clinic of the University of Athens where mechanical ventilation and air purifying systems were operated during working shifts. The main scope of this experiment was to examine the levels of IAQ within the room along with the impact of the air purifiers on selected air pollutants. The most important conclusions drawn are the following:During working days, thermal comfort was relatively satisfactory with slightly increased values of indoor temperature for a few cases and the majority of RH values were within the proposed limit range.The use of CO_2_ as an indicator for the ventilation levels of the clinic led to the conclusion that the mechanical ventilation system retained adequate levels within the room even when the human presence indoors was high.The levels of TVOC measured were relatively low even if many instant maximums were recorded due to the frequent use of dental materials within the clinic during the working shifts.PM_2.5_ and PM_10_ were significantly decreased for the duration of the experimental period with a few exceptions, mainly due to external atmospheric pollution episodes.The use of mechanical ventilation during all working shifts played a key role in the results, retaining good IAQ levels within the clinic.Air purifiers showed a positive impact on IAQ, mainly by reducing the levels of PM_2.5_ and secondly of TVOC.P2 was found to be more effective than P1 for all examined air pollutants.

## Figures and Tables

**Figure 1 ijerph-18-08886-f001:**
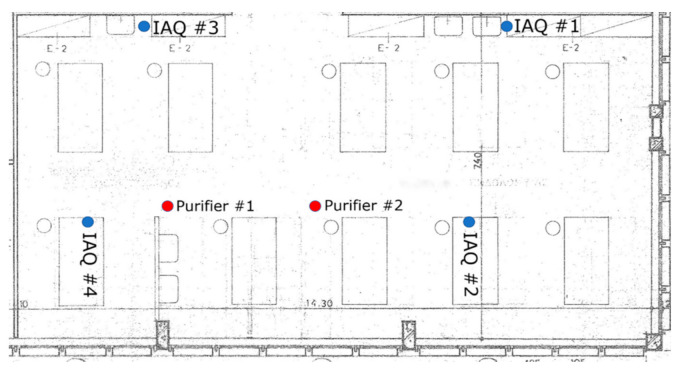
Positions of the IAQ sensors and the air purifiers in the clinic.

**Figure 2 ijerph-18-08886-f002:**
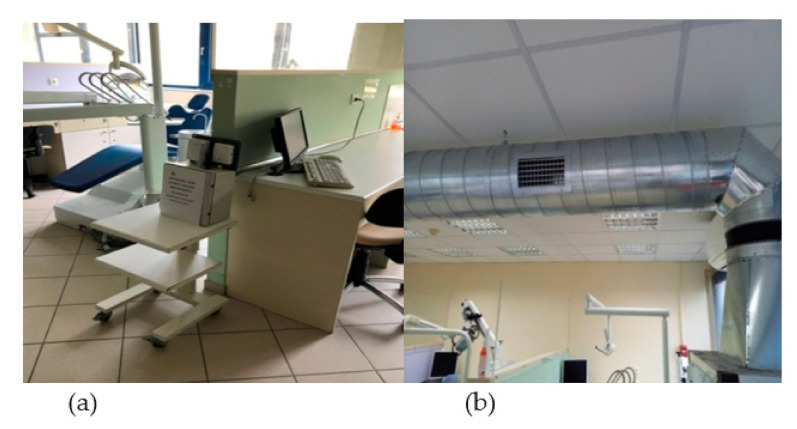
A view of (**a**) an IAQ Tongdy TSP-18 sensor and (**b**) the mechanical ventilation system in the clinic.

**Figure 3 ijerph-18-08886-f003:**
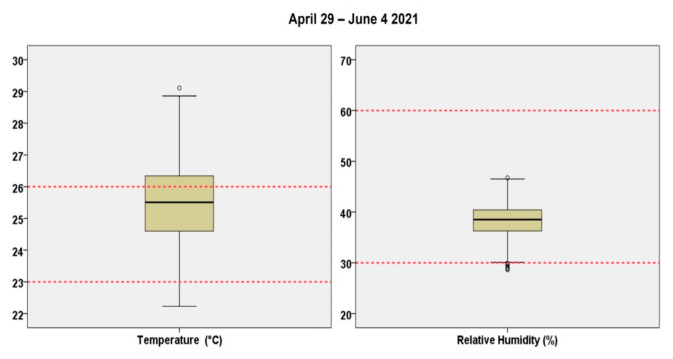
Levels of temperature (°C) and relative humidity (%) during the working days in the clinic, 29 April–4 June 2021.

**Figure 4 ijerph-18-08886-f004:**
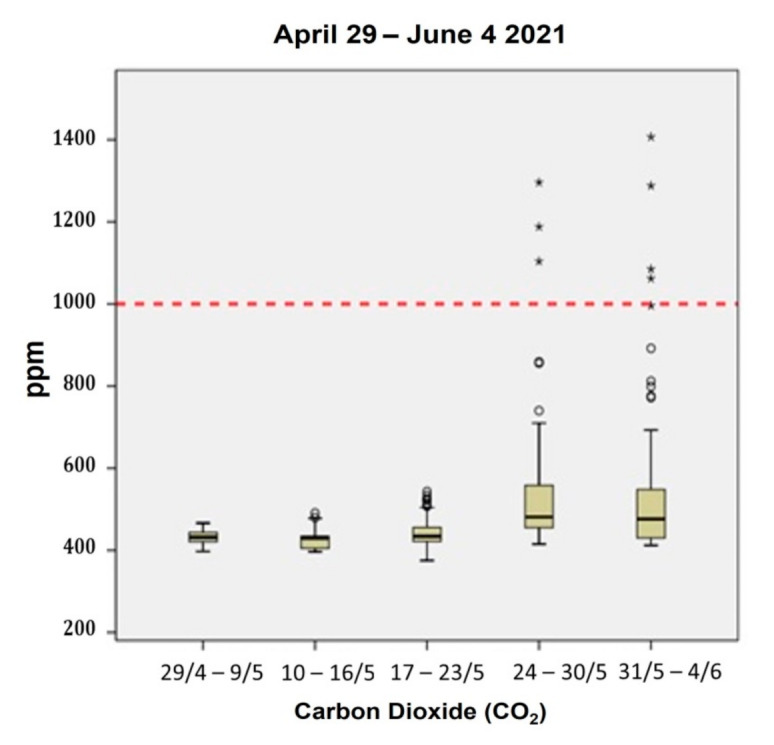
Concentrations of CO_2_ (ppm) in the clinic per monitoring period, 29 April–4 June 2021. The symbol “*” illustrates extreme values.

**Figure 5 ijerph-18-08886-f005:**
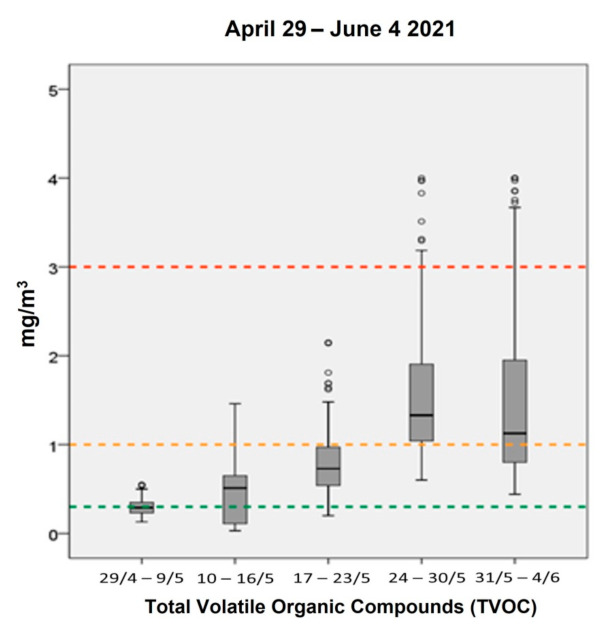
Concentrations of TVOC (mg/m^3^) in the clinic per monitoring period, 29 April–4 June 2021.

**Figure 6 ijerph-18-08886-f006:**
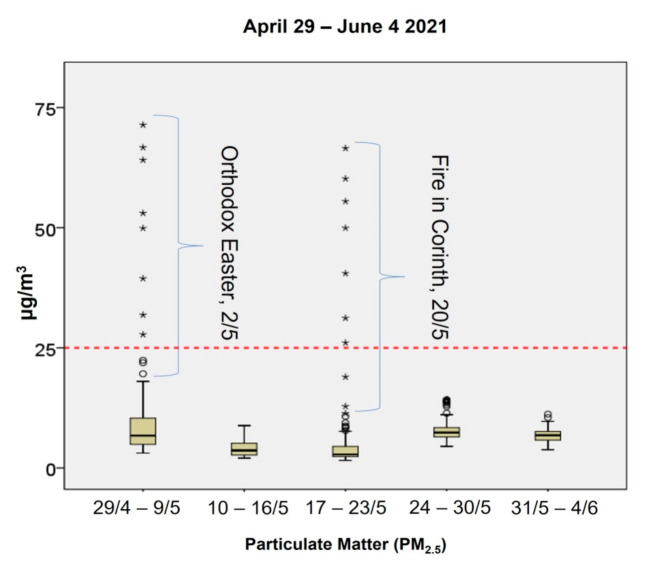
Concentrations of PM_2.5_ (μg/m^3^) in the clinic per monitoring period, 29 April–4 June 2021. The symbol “*” illustrates extreme values.

**Figure 7 ijerph-18-08886-f007:**
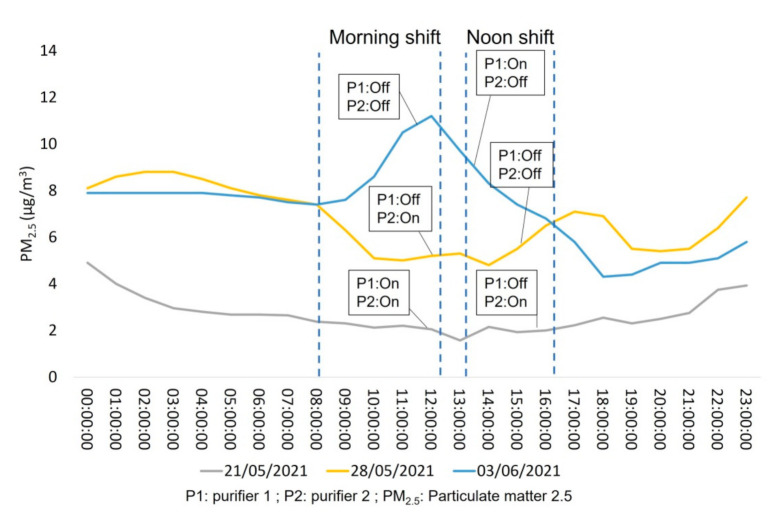
Hourly distribution of PM_2.5_ during three days of interest under different operational patterns of air purifiers.

**Figure 8 ijerph-18-08886-f008:**
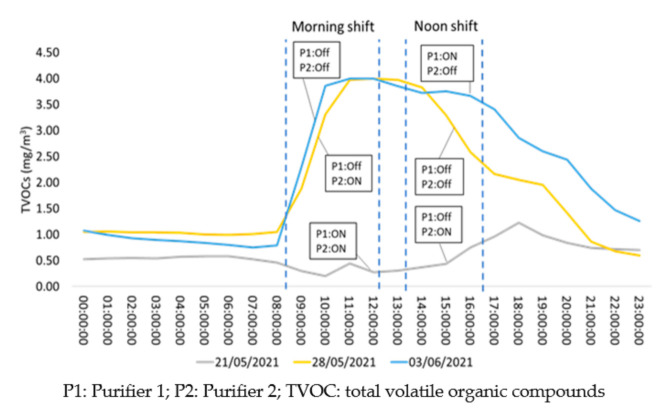
Hourly distribution of TVOC during three days of interest under different operational patterns of air purifiers.

**Table 1 ijerph-18-08886-t001:** Occupancy levels and operational patterns of the two air purifiers of the clinic per monitoring period, 17 May–4 June 2021.

Monitoring Period	Shifts	Occupancy (Number of People)	Mechanical Ventilation	Purifier 1	Purifier 2
17–21 May 2021	08:30 a.m.–12:45 p.m.	27	On	On	On
	13:30–16:30 p.m.	16	On	Off	On
24–28 May 2021	08:30 a.m.–12:45 p.m.	32	On	Off	On
	13:30–16:30 p.m.	26	On	Off	Off
31 May–4 June 2021	08:30 a.m.–12:45 p.m.	35	On	Off	Off
	13:30–16:30 p.m.	29	On	On	Off

**Table 2 ijerph-18-08886-t002:** Descriptive statistics for all investigated parameters, 29 April–4 June 2021.

Descriptive Statistics	Relative Humidity (%)	Temperature (°C)	TVOC (mg/m^3^)	CO_2_ (ppm)	PM_10_ (μg/m^3^)	PM_2.5_ (μg/m^3^)
Mean	39.49	25.76	0.79	457	6.80	6.30
Median	39.74	25.69	0.58	430	5.63	5.10
Standard Deviation	4.55	1.57	0.76	102	6.71	6.63
Minimum	26.55	21.45	0.03	375	2.05	1.58
Maximum	51.79	30.08	4.00	1407	72.63	71.40

TVOC: total volatile organic compounds; CO_2_: carbon dioxide; PM_10_ and PM_2.5_: particulate matter <10 micrometers and <2.5 micrometers, respectively.

**Table 3 ijerph-18-08886-t003:** Increasing rates of particulate matter and TVOC for different operational patterns of air purifiers during the morning shifts.

**Air Pollutants**	17–21 May 2021 All Systems: On	**24–28 May 2021**	**31 May–4 June 2021**
		V:On|P1:Off|P2:On	V:On|P1:Off|P2:Off
PM_10_		40%	49%
PM_2.5_		53%	64%
TVOC		170%	151%

V: mechanical ventilation system; P1: purifier 1; P2: purifier 2; PM_2.5:_ particulate matter 2.5; PM_10_: particulate matter 10; TVOC: total volatile organic compounds.

**Table 4 ijerph-18-08886-t004:** Increasing rates of particulate matter and TVOC for different operational patterns of air purifiers during the noon shifts.

**Air Pollutants**	17–21 May 2021 V:On P1:Off P2:On	**24–28 May 2021**	**31 May–4 June 2021**
		V:On|P1:Off|P2:Off	V:On|P1:On|P2:Off
PM_10_		170%	129%
PM_2.5_		202%	153%
TVOC		226%	234%

V: mechanical ventilation system; P1: purifier 1; P2: purifier 2; PM_2.5:_ particulate matter 2.5; PM_10_: particulate matter 10; TVOC: total volatile organic compounds.

## Data Availability

Data are available upon permission request from the Ethics Committee of the Dentistry School.

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
