# Peer review of "Indoor Air Quality Evaluation Using Mechanical Ventilation and Portable Air Purifiers in an Academic Dentistry Clinic during the COVID-19 Pandemic in Greece"

_ijerph, 2021, doi:10.3390/ijerph18168886_

Round 1
Reviewer 1 Report
In this research, authors tried to assess the air quality in the dental hospital facility during the various phases of COVID-19 pandemic. In my opinion, this work is presented correctly and I recommend publishing the article. However, prior to that, I would like to address some issues that I wish from the authors to be corrected in the final version.
- There are several typing errors throughout the text. For example, in line 39 PM2.5 is written PM 2.5, PM10 as PM 10 pm, and CO2 as CO2. In line 123 "2 m" is written as "2m", etc. I urge the authors to check all of these errors and correct them.
- In the introduction, authors claim that COVID-19 pandemic is unprecedented (line 48) which is certainly not true. There were some pandemics in the history that lasted for decades. Even at this moment, HIV pandemic lasts for 4 decades. I would recommend rephrasing or omitting this part of the sentence.
- Figure description for figures 3 to 6 are very scarce and not completely informative. Since figures have to be self-explenatory, I would suggest more detailed description of what is on them. Also, on figures 5 and 6 TVOC and PM2.5 are described twice (as part of the figure and later in the figure description). This should be corrected in the way that description on the figure should be removed.
- Table 2 should be placed on the single page.
Author Response
We thank the Reviewer for the valuable comments in order to improve our manuscript.
- We corrected the errors throughout the text, as recommended.
- We deleted the word “unprecedented” as recommended.
- We provided more details in Figures 3 to 6, as recommended. We also corrected Figures 5 and 6 description, as suggested by the reviewer.
- As requested, we submitted Table 2 on a single page as well.
Reviewer 2 Report
The manuscript details a study on the effect of ventilation and air purifiers on indoor air quality, especially on PM2.5 which can play a role in the transmission of Covid-19 virus, in a clinic of a dentistry school of an university in Athens. Basically the results show that ventilation and air purifiers reduce the level of pollutants such as CO2, PM2.5, PM10 and TVOC (total VOC) after accounting for some events and activities that can cause high PM and VOC in an dental environment. The results are expected and specific to the settings of this clinic. I am not sure that it can be applied to other environment settings at other location. It would be useful if the authors can quantify how much improvement overall in the concentration of each of the pollutants from using ventilation and air purifiers. The sensors used in the study should also be described in more details.
There ae other minor corrections that should be addressed
(1) Line 39: "PM 2.5, PM 10 pm" should be "PM2.5, PM10"
(2) Line 141: TSP-18sensors should be TSP-18 sensors
(3) Line 161: "...the threshold prices of CO2,.." should be "... the threshold concentration of CO2,..."
(4) Line 185: "particulate matter 10 & 2.5" should be "parti
culate matter < 10 micrometer & <2.5 micrometer"
(5) Line 279: "(form each operational week)" should be "(from each operational week)"
(6) Line 285: "In 3/6 when.." should be "On 3 June when...". Rather than using the date as 3/6, the authors should specify clearly the day and month. This should be applied in line 290, 291...
(7) Line 289: "The behavior of TVOC 289 found to be.." should be "The behaviour of TVOC 289 is found to be..."
Author Response
We thank the Reviewer for the valuable comments in order to improve our manuscript. As requested, we checked English language and style, and spell checked our manuscript.
Answer to major comment
The results of our research are expected and specific to the settings of this clinic. We cannot be sure that they can be applied to other environment settings at other locations. We have to state that undoubtedly every location has different architectural, dimensional and operational characteristics. The environmental characteristics of a supermarket are different to the characteristics of an airport and the characteristics of healthcare facilities are different to the characteristics of a residential construction, due to the different demands and emissions from every activity. As requested, additional details regarding the performance of the Tongdby sensors have been added.
Answer to minor comments
1.We corrected as requested.
2.We corrected, as requested.
- We corrected, as requested.
- We corrected, as requested.
- We corrected, as requested.
- We corrected, as requested throughout the text.
- We corrected, as requested.
Reviewer 3 Report
This study investigated the effect of air purifier on indoor air quality over a dental facility, especially on PM2.5 which can potentially help in promoting the transmission of the novel virus SARS-CoV-2. The result shows that the air purifiers remove the indoor air pollutants efficiently. The paper is well written and laid out, telling a convincing story. I would recommend publication if the authors could address the following minor comments.
Minor comments:
- As suggested by the authors, the CO2 is an indicator for ventilation levels of clinic. It would be interesting to see the change of pollutant/CO2 ratio in the experiment period to separate the impact of purifiers and ventilation. Another way to show it is to have additional figure showing the diurnal cycle of CO2 concentrations in the days of interest.
- The case study is quite interesting. However, using a single day in each period loses generality. Can the authors show the averaged diurnal profile over the whole period and then compare the averaged profile?
- Since the mechanical ventilation system is always on, it is not possible to reach the conclusion that the ventilation system plays a key role in retaining the CO2 level.
Other comments:
Line 39: please use consistent notation (subscript) for PM2.5, PM10, and CO2
Line 75: “In” the past decade
Line 206: the “significant” should associate with statistical tests. Consider change it to other words such as “dramatically” or “largely” or indicate the p value.
Author Response
We thank the Reviewer for the valuable comments in order to improve our manuscript. As requested, we checked English language and style, and spell checked our manuscript.
Answer to Reviewers comments
1.Because of ethical issues, it is not feasible to turn off the mechanical ventilation system. In Figure 4 the levels of CO2 are depicted per monitoring period (four in total) throughout the study period (April 29 through June 4, 2021).
2.In Figure 4 the levels of CO2 are shown per monitoring period. In particular, these values do not represent single-day values in each monitoring period, but rather the values throughout each monitoring period.
- We agree with the reviewer, however because of ethical issues, it is not feasible to turn off the mechanical ventilation system.
Answer to other comments
1.Line 39: We corrected throughout the text, as recommended.
2.Line 79: We added the word “In”, as recommended.
- Line 206: we rephrased to “largely” as recommended.